# Analysis of leaf morphology development-related genes and photosynthetic metabolic pathways in the transcriptomes of new leaves of Tea-Oil tree (*Camellia oleifera* 'changlin53')

**Zongshun Zhou**, **Chuansong Chen**, **Ying Jiang**, **Yikai Hua**, **Yisi Liu**, **Hang Wang**, **Lixian Cao**, **Shuhua Wu**, **Hongyan Guo**\*

Jianxi Provincianl Key Laboratory of Cultivation and Utilization of Camellia Sinensis Resources, Experimental Center of Subtropical Forestry, Chinese Academy of Forestry, Fenyi, Jiangxi, China

\* guohongyan5@caf.ac.cn

## Abstract

*Camellia oleifera* is a distinctive woody oilseed species endemic to southern China and is considered one of the world's four major woody oil plants alongside oil palm, olive, and coconut. Its oil is renowned for its high content of unsaturated fatty acids, earning it the title "Oriental Olive Oil." However, the conventional *C. oleifera* industry faces challenges such as substantial yield fluctuations and poorly understood regulatory mechanisms of oil metabolism. There is an urgent need to elucidate the synergistic mechanisms between photosynthetic efficiency and metabolic networks at the molecular level during leaf development to establish a theoretical foundation for high-yield, high-quality breeding. In this study, we systematically analyzed the developmental dynamics of new leaves at the spreading, growth, and maturity stages in the high-yield variety 'Changlin 53'. By integrating morphological and anatomical observations with multi-period transcriptome sequencing, we investigated the key regulatory networks underlying leaf functional maturation. The results showed that as leaves developed, chlorophyll *a* content increased significantly, the expression of photosynthesis-related genes (e.g., chlorophyll *a*-binding protein Lhca1, photosystem II reaction center protein PsbA) was upregulated, photosynthetic capacity was gradually enhanced, and leaf functional maturation was promoted through the regulation of carbon assimilation and energy metabolism pathways. To validate these findings, 12 randomly selected differentially expressed genes (DEGs) were analyzed using qRT-PCR. The expression patterns were highly consistent with the RNA-Seq data. This study reveals, for the first time, the potential synergistic mechanism involving photosynthesis, metabolism, and hormones during leaf development in 'Changlin 53', providing a key theoretical basis for selecting and breeding *C. oleifera* varieties with high light-use efficiency.

**Data availability statement:** All relevant data are within the manuscript and its Supporting Information files.

**Funding:** This research was funded by Research on Efficient Propagation of High-Quality Tea Seed Varieties and Establishment Techniques for Cutting Nurseries, grant number JXYCZX〔2023〕114, Research on Breeding and Application of High-Quality Red Flower Tea Tree Varieties, grant number YCYJZX〔2023〕131, and Investigation and Collection of Germplasm Resources for Drought-Tolerant Non-wood Forest in the Belt and Road , grant number（CAFYBB2025GC002）. The funders had no role in study design, data collection and analysis, decision to publish, or preparation of the manuscript.

**Competing interests:** The authors have declared that no competing interests exist.

## Introduction

As the primary organs for photosynthesis and material metabolism, leaf morphology and functional development directly determine the efficiency of light energy capture and the accumulation of assimilates, serving as pivotal regulatory nodes in crop yield and quality formation [1]. Leaf development is coordinately regulated by genetic networks [2], hormone signaling [3], and environmental responses [4]. Among these, the spatiotemporal coupling of carbon/nitrogen metabolism with photosynthesis is particularly critical. Research has shown that dynamic regulation of chloroplast photosystem assembly efficiency and Calvin cycle enzyme activity significantly affects leaf carbon assimilation flux. The metabolic network maximizes photosynthetic efficiency by optimizing the allocation of light reaction products (ATP/NADPH) to dark reaction substrates (RuBP) [5]. Therefore, analyzing the synergistic regulation of metabolism and photosynthesis during leaf development holds great theoretical value for breeding high-yield, high-quality crops.

The advent of high-throughput sequencing has made RNA-Seq a pivotal tool for elucidating molecular mechanisms in leaf development [6,7]. In model plants, this technology has revealed the role of the GT-2 transcription factor family in regulating tomato leaf morphology by modulating cell expansion-related genes (e.g., EXPANSIN) [8]. Furthermore, transcriptome dynamics during key maize leaf development stages have been systematically characterized, highlighting intricate hormonal interactions (e.g., IAA-CTK homeostasis) [9]. Among woody economic species, *C. oleifera* has been recognized by the FAO as a healthy edible oil source and integrated into China's bulk oilseed industry due to its oil rich in unsaturated fatty acids and bioactive components [10]. The entire plant is valuable in food, medicine, and ecological restoration [11,12]. However, growing demand for high-quality edible oil has led to a supply-demand imbalance. Research demonstrates that leaves are the main organs for photosynthetic carbon sequestration, and their photosynthetic capacity, nutrient transport efficiency, and physiological status are closely related to plant yield [13]. For instance, tomato varieties with round leaves ('Stupice', 'Glacier') show higher fruit soluble solids content and yield compared to pinnately lobed varieties [14]. Therefore, studying the leaf development regulatory network in *C. oleifera* is essential for improving its quality and productivity. However, most previous studies focused on mature leaf physiology [15,16]. leaving a gap in systematic knowledge about gene expression dynamics and photosynthesis-metabolism synergy during early leaf development.

In this study, we analyzed gene expression in fresh leaves of *C. oleifera* 'Changlin 53' at three developmental stages based on comprehensive morphological observations and chlorophyll content measurements using RNA-Seq. Our goal was to lay a foundation for understanding the molecular mechanisms and regulatory networks governing tea-oil tree leaf development.

## Materials and methods

### Plant growth and sample collection

Twelve-year-old *C. oleiferaa* ('Changlin53') plants with uniform growth and no disease were grown under natural conditions at the Experimental Center for Subtropical

Forestry, Chinese Academy of Forestry (Fenyi County, Jiangxi, China; 27°49' N, 114°39' E; altitude 88–92 m). Standard fertilizer and water management were applied. From 12 February to 4 June 2019, bud and new leaf samples were collected every 7 days for morphological analysis. Based on morphological results [17], hree key stages were selected: sprouting (Stage I, NLIII), flower bud primordium formation (Stage II, NLIV), and petal differentiation (Stage III, NLV). After sampling, tissues were snap-frozen in liquid nitrogen and stored at –80 °C until RNA extraction. Three biological replicates were used per stage, each consisting of a mix from three plants.

### Analysis of morphological characteristics

Before each sampling, experimental materials were photographed with a camera and a stereomicroscope. Buds were fixed in FAA (5% formaldehyde, 5% acetic acid, 63% anhydrous ethanol, 27% pure water). Samples were dehydrated through an ethanol series (70% twice, 80%, 90%, 95% once each, 100% twice; 30 min each), dried using a critical point dryer (Hitachi HCP-2) with liquid $CO_2$, and gold-coated with an ion sputter coater (Edwards E-1010).

### Chlorophyll content measurement

Chlorophyll was extracted from fresh leaf tissue (0.1 g) using 80% acetone in the dark. Absorbance of the supernatant was measured at 663 nm, 645 nm, and 470 nm using a spectrophotometer. Chlorophyll *a*, chlorophyll *b*, and carotenoid concentrations were calculated using Arnon's equations ^[Arnon1949].

### RNA extraction, library construction, and sequencing

Total RNA was extracted using the RNA prep Pure DP441 Kit (Tiangen, Beijing) following the manufacturer's protocol. RNA integrity was checked via agarose gel electrophoresis, purity with a NanoDrop spectrophotometer, concentration with a Qubit 2.0 Fluorometer, and integrity number (RIN) with an Agilent 2100 Bioanalyzer. High-quality RNA samples (RIN > 7.0) were used for library construction. Poly(A) mRNA was enriched using oligo(dT) beads, fragmented, and reverse-transcribed into cDNA. After end repair, A-tailing, and adapter ligation, cDNA libraries were amplified by PCR. Libraries were quantified and pooled for paired-end 150 bp sequencing on an Illumina NovaSeq 6000 platform by Gene Denovo Biotechnology Co., Ltd. (Guangzhou, China).

### Transcriptome data processing and De Novo assembly

Raw reads were filtered to remove adapters and low-quality bases (Q < 20) using fastp v0.23.2. Clean reads from all nine samples were pooled for de novo assembly using Trinity v2.8.4 (kmer_size = 31, min_kmer_cov = 17, normalize_max_read_cov = 50) [18]. The longest transcript per gene was selected as a unigene. Assembly quality was assessed with BUSCO. Unigene expression was quantified as FPKM (Fragments Per Kilobase per Million mapped reads) using RSEM [19]. Functional annotation was performed against Nr, Swiss-Prot, Pfam, COG/KOG, KEGG, and GO databases using Diamond BLASTx (E-value ≤ 1e-5).

### Differential expression and enrichment analysis

DEGs between stages (NLIII vs. NLIV, NLIII vs. NLV, NLIV vs. NLV) were identified using DESeq2 in R with thresholds |log$_2$ (fold change)| > 1 and FDR < 0.05. GO and KEGG enrichment analyses were performed using clusterProfiler, with terms/pathways considered enriched at FDR < 0.05. Visualization was done with ggplot2 and pathview.

### Weighted Gene Co-expression Network Analysis (WGCNA)

A weighted gene co-expression network was constructed using the WGCNA R package to identify gene modules associated with leaf developmental stages. A soft-thresholding power of 8 was selected to achieve a scale-free network topology.

 

Genes were clustered into modules using a minimum module size of 50 and a merge cut height of 0.25. Module eigengenes (MEs) were calculated and correlated with the developmental trait (stages NLIII, NLIV, NLV) to identify significant module-trait relationships. Hub genes within key modules were defined by high module membership (MM > 0.8) and gene significance (GS > 0.2). Functional enrichment analysis of module genes was performed for GO terms and KEGG pathways.

### GO and KEGG enrichment analysis of DEGs

KEGG and GO enrichment analyses were conducted to investigate pathways and biological functions that were activated during flower transition in *C. oleifera*. The background documents of GO and KEGG were from GO and KEGG annotation. GO and KEGG enrichment analyses of DEGs were performed using OmicShare tools in 2021, a free online platform for data analysis (http://www.omicshare.com/tools, 16 August 2022).

### Quantitative Real-Time PCR (qRT-PCR) validation

Total RNA (1 μg) from the same samples used for RNA-Seq was reverse-transcribed using the Prime Script RT reagent Kit with gDNA Eraser (Takara). qRT-PCR was performed with TB Green Premix Ex Taq II (Takara) on a Quant Studio 6 Flex system. The *C. oleifera* Actin gene (CoActin) was used as an internal control. Relative expression was calculated using the $2^{-\Delta\Delta Ct}$ method [20]. Three biological and three technical replicates were performed. Data are presented as mean ± SD. Statistical significance was determined by one-way ANOVA followed by Tukey's HSD test (p < 0.05). Primers are listed in Supplementary S1 Table.

### Statistical analysis

All physiological data are presented as mean ± standard error (SE) of three biological replicates. Differences among stages were assessed by one-way ANOVA with Tukey's test (p < 0.05) using SPSS 22.0.

## Results

### Morphological and physiological dynamics during new leaf development

New shoots of *C. oleifera* originated from dormant buds formed in previous growth cycles. Longitudinal observation from 12 February to 26 March 2019 revealed three phases: dormancy (12 Feb – 12 Mar), bud elongation and tip conification (from 19 Mar), and shoot differentiation with bract emergence (26 Mar) (Fig 1). Subsequent shoot growth showed three phases: rapid elongation and coloration (brown to green, 2–16 Apr), slow growth and stem maturation (23 Apr – 21 May), and completion of shoot development with dark green leaves (from 28 May) (Fig 2A, B).

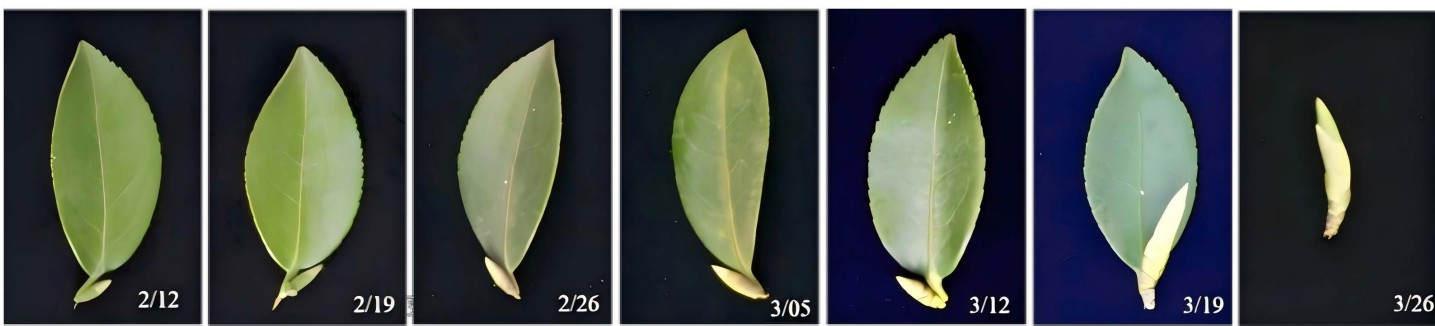

**Fig 1. Investigation on leaf bud growth of *C. oleifera*.**

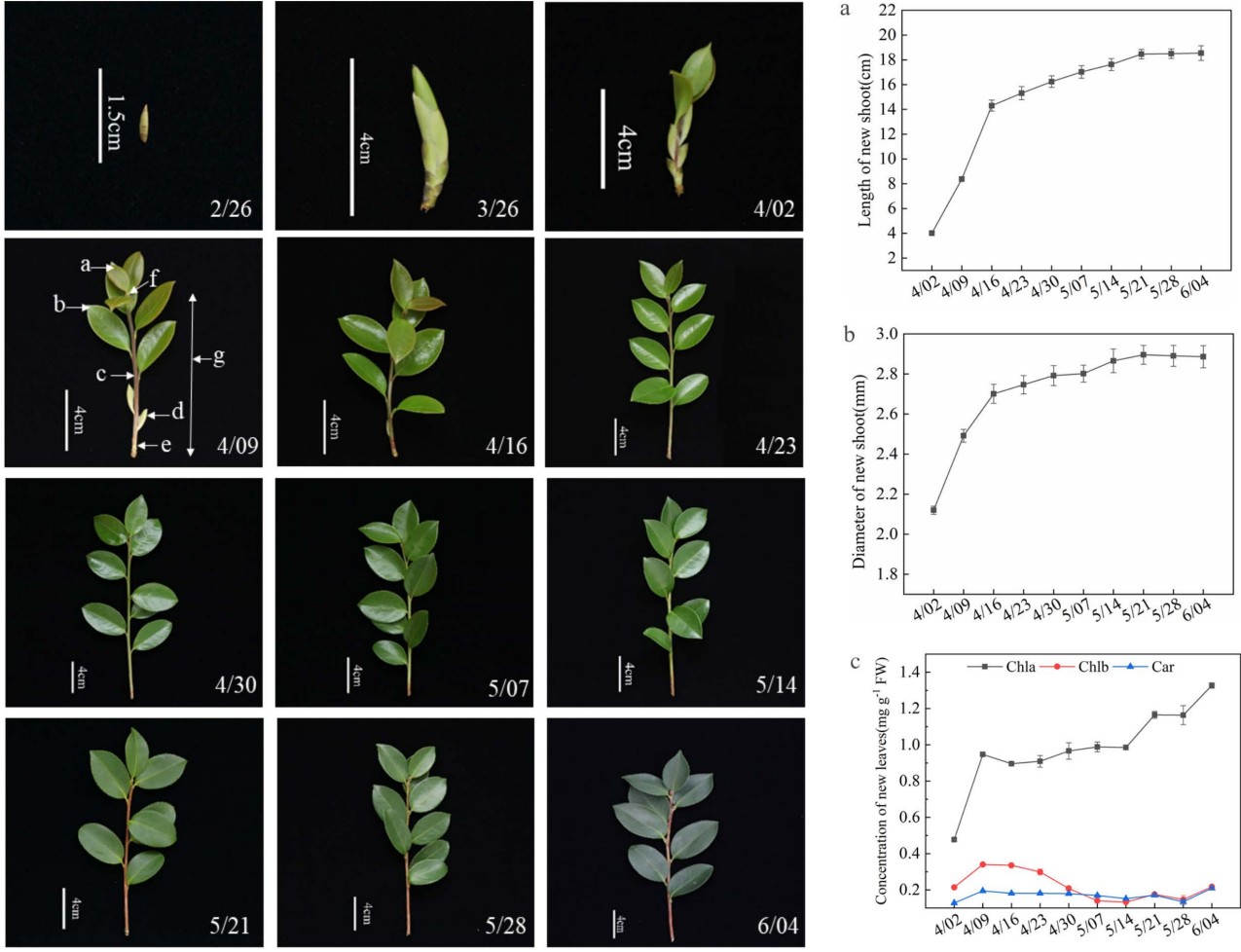

**Fig 2. A: Investigation on new shoots growth of *C. oleifera*, they should be listed as: (a) apical leaf, (b) new leaves,(c) new shoot, (d) bract, (e) position of new shoot thickness measurement, (f)apical bud, (g) length of new shoot. B:** Length (a) and diameter (b) of new shoots during growth,they should be listed as: Bars indicate mean ± standard error (SE). The same below.(c): Chlorophyll concentration of new leaves during the new shoots growth.

## Sequencing quality analysis

RNA-Seq of nine samples yielded 37–53 million clean reads per sample, with Q20 > 97.3%, Q30 > 92.3%, and GC content 45.9–52.9% (Table 1), indicating high-quality data. De novo assembly generated 120,081 unigenes (N50 = 23,906 bp, average length = 703 bp). Over 58% of unigenes were annotated in public databases (Nr: 59,793; Swiss-Prot: 38,030; KEGG: 52,785). Principal component analysis (PCA) clearly separated samples by developmental stage (Fig 3), and biological replicates showed high correlation ($R^2 > 0.95$) (Fig 4), confirming sample specificity and data reliability.

## Identification of differentially expressed genes and key transcription factors

Pairwise comparisons identified 8,225 DEGs between NLIII and NLIV, 9,443 between NLIII and NLV, and 981 between NLIV and NLV (Fig 5A). A total of 11,317 unique DEGs were obtained (Fig 5B). Notably, the number of DEGs decreased markedly from NLIV to NLV, suggesting a transition from active morphogenesis to functional stabilization.

**Table 1. Overview of the samples for RNA-seq and the number of clean reads.**

| Tissues | Sample name | Number of clean reads | Number of clean data | Q20(%) | Q30(%) | GC(%) |
|---|---|---|---|---|---|---|
| New leaves | NLIII-1 | 42837560 | 6356285735 | 6192299676 (97.42%) | 5888479271 (92.64%) | 3001478960 (47.22%) |
| | NLIII-2 | 40528648 | 6005598645 | 5846774084 (97.36%) | 5550529822 (92.42%) | 2827134711 (47.07%) |
| | NLIII-3 | 42894172 | 6387408955 | 6227933348 (97.50%) | 5922689515 (92.72%) | 3003378142 (47.02%) |
| | NLIV-1 | 53415014 | 7970972745 | 7815364603 (98.05%) | 7508787039 (94.20%) | 3908530068 (49.03%) |
| | NLIV-2 | 45737150 | 6814526366 | 6679283533 (98.02%) | 6412626697 (94.10%) | 3443590528 (50.53%) |
| | NLIV-3 | 47382172 | 7085119589 | 6956544847 (98.19%) | 6699319077 (94.55%) | 3536178759 (49.91%) |
| | NLV-1 | 39201664 | 5851229085 | 5692570819 (97.29%) | 5400283193 (92.29%) | 2904043037 (49.63%) |
| | NLV-2 | 46401330 | 6917175120 | 6732268703 (97.33%) | 6393404064 (92.43%) | 3552745886 (51.36%) |
| | NLV-3 | 39292826 | 5866041557 | 5708713349 (97.32%) | 5417028820 (92.35%) | 2963404446 (50.52%) |

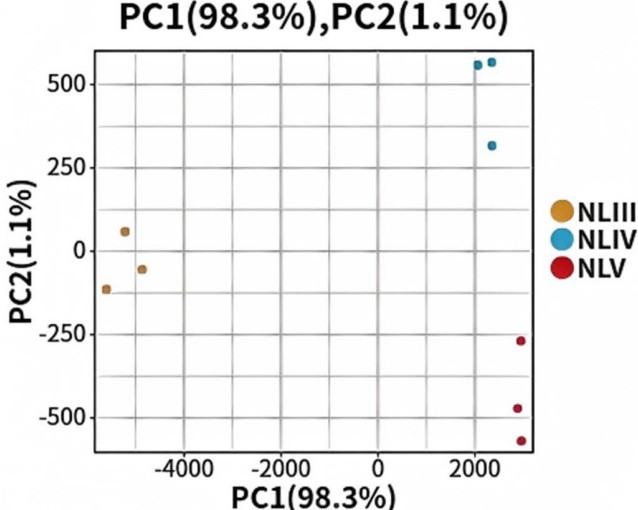

**Fig 3. Principal component analysis of the RNA-seq data in all samples.**

Among the DEGs, 1,214 were annotated as transcription factors (TFs). MYB, bHLH, NAC, ERF, and WRKY families were predominant (Supplementary S1 Fig). Many TFs showed stage-specific expression; e.g., several NAC TFs were upregulated from NLIII to NLIV, potentially regulating cell expansion and secondary wall formation, while a subset of WRKY TFs was induced at NLV, possibly involved in stress response and maturation.

### GO enrichment analysis of DEGs

GO enrichment analysis revealed distinct biological processes across comparisons (Fig 6). For NLIII vs. NLIV, terms related to "precursor metabolite and energy generation," "cell wall biogenesis," and "response to external stimulus" were top enriched (Fig 7A), indicating active metabolic building and stress acclimation during early expansion. In NLIII vs. NLV, terms shifted to "hormone transport," "sugar metabolism," "vascular development," and "meristem regulation" (Fig 7B), aligning with the transition to functional maturity and source-sink establishment.

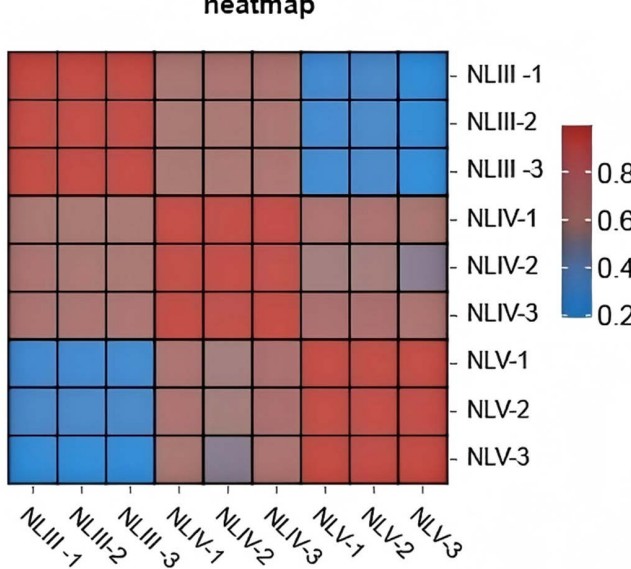

**Fig 4. Correlation analysis of the RNA-seq data in all samples.**

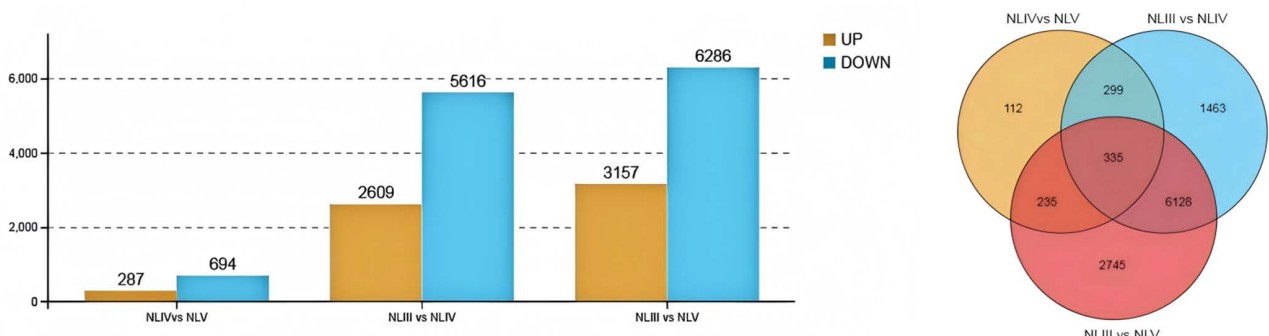

**Fig 5. A: Up and down regulated DEGs in each comparison of new leaves; B: Venn diagram showing the DEGs on distinct comparisons of new leaves.**

## KEGG enrichment of DEGs

In order to characterize the function of DEGs screened in the metabolic pathways in each comparison. KEGG pathway enrichment analysis was performed on these DEGs. The results indicated that the 1950 annotated DEGs were enriched into 129 KEGG pathways, which could be further categorized into 19 class B KEGG pathways and 5 class A KEGG pathways, As illustrated in Figs 8, the DEGs enriched into the class A KEGG pathway "Metabolism" (Metabolism) of the class B KEGG pathway and DEGs were the most abundant.

We analyzed the DEGs of each group and found that 1871 annotated DEGs enriched into 129 KEGG pathways, of which 30 pathways were significantly enriched (P<0.05) and 20 pathways were highly significantly enriched (P<0.01). We found that the top 20 significantly enriched KEGG pathways were mainly involved in metabolic pathways, flavonoid biosynthesis, phenylpropane-like biosynthesis, phytohormone signaling, biosynthesis of secondary metabolites, biosynthesis

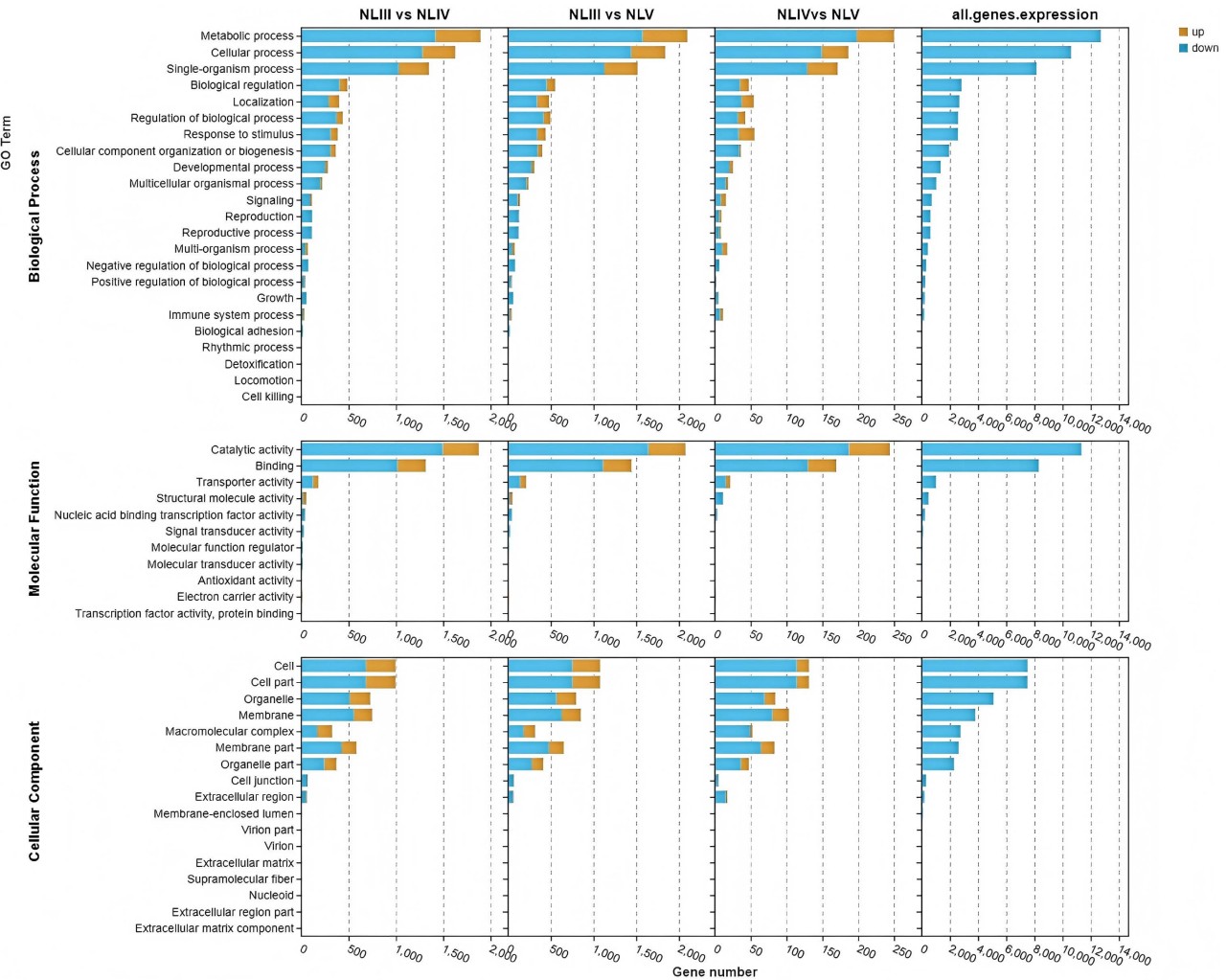

**Fig 6. GO function analysis of DEGs in each comparison of new leaves.**

of stilbenes, diarylheptanes and gingerols, photosynthesis, steroid biosynthesis, linoleic acid metabolism, fatty acid elongation, pentoses and glucose acid ester interconversion, aminosugar and nucleotide sugar metabolism, fatty acid biosynthesis, oxidative phosphorylation, monomycin biosynthesis, terpene polymeric molecule backbone biosynthesis, o-glycan biosynthesis of their tower types, sesquiterpene and triterpene biosynthesis, fatty acid metabolism, and galactose metabolism. The results indicate that in addition to biosynthesis, phytohormone signaling plays a role and affects the efficiency of photosynthesis and the development of photosynthetic organs in plants by regulating growth and metabolism.

Analysis of the DEGs of NLIV vs NLV revealed that the annotated 314 DEGs were enriched into 88 KEGG pathways, of which 13 KEGG pathways were significantly enriched ($P < 0.05$) and 8 KEGG pathways were highly significantly enriched ($P < 0.01$). Analysis of the top 20 significantly enriched KEGG pathways revealed that they were mainly involved in photosynthesis, phenylpropane-like biosynthesis, photosynthesis-haptoglobin, unsaturated fatty acid biosynthesis, metabolic pathways, fatty acid elongation, fatty acid metabolism, carbon fixation in biological photosynthesis, secondary metabolite biosynthesis, biosynthesis of stilbene, diarylheptane, and gingerol synthesis, phytohormone signaling, flavonoid biosynthesis, biosynthesis of sesquiterpenes and triterpenes, biosynthesis of keratins, seabed alkaloids, and waxes, biosynthesis of ubiquinone and other terpene quinones, carbon metabolism, glycosphingolipid biosynthesis- lactic acid

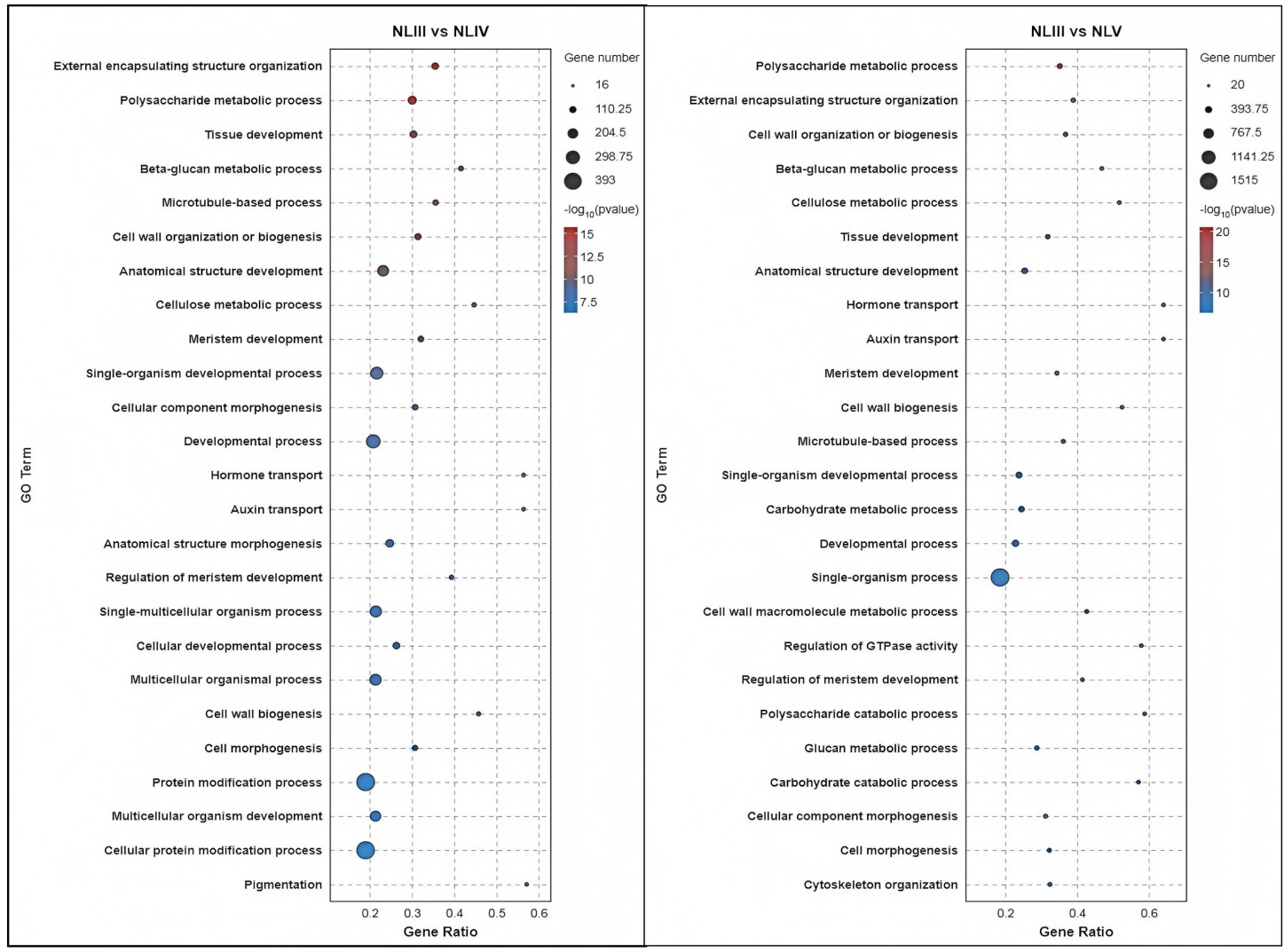

**Fig 7. Top 25 significance level GO terms in each comparison of new leave.**

and lactose series, diterpene biosynthesis, fatty acid degradation, and nitrogen metabolism (Figs 9). The results indicated that gene expression in the NLIV to NLV stages was centered on consolidating photosynthetic capacity, strengthening secondary metabolic defenses, and perfecting the lipid barrier, driving leaf functional maturation and environmental adaptation.

## WGCNA identifies a photosynthesis-related module linked to leaf maturation

WGCNA of the transcriptome data grouped genes into 10 co-expression modules. The turquoise module showed the strongest significant correlation with leaf development, exhibiting a marked negative correlation (r = −0.92, p = 2e-5) with progression from the young (NLIII) to mature (NLV) stage.

Hub gene analysis within the turquoise module identified genes with high intramodular connectivity, such as Unigene0000022 and Unigene0000021 (MM > 0.99). The strong correlation between gene significance (GS) for development and module membership (MM) confirmed the central role of these genes.

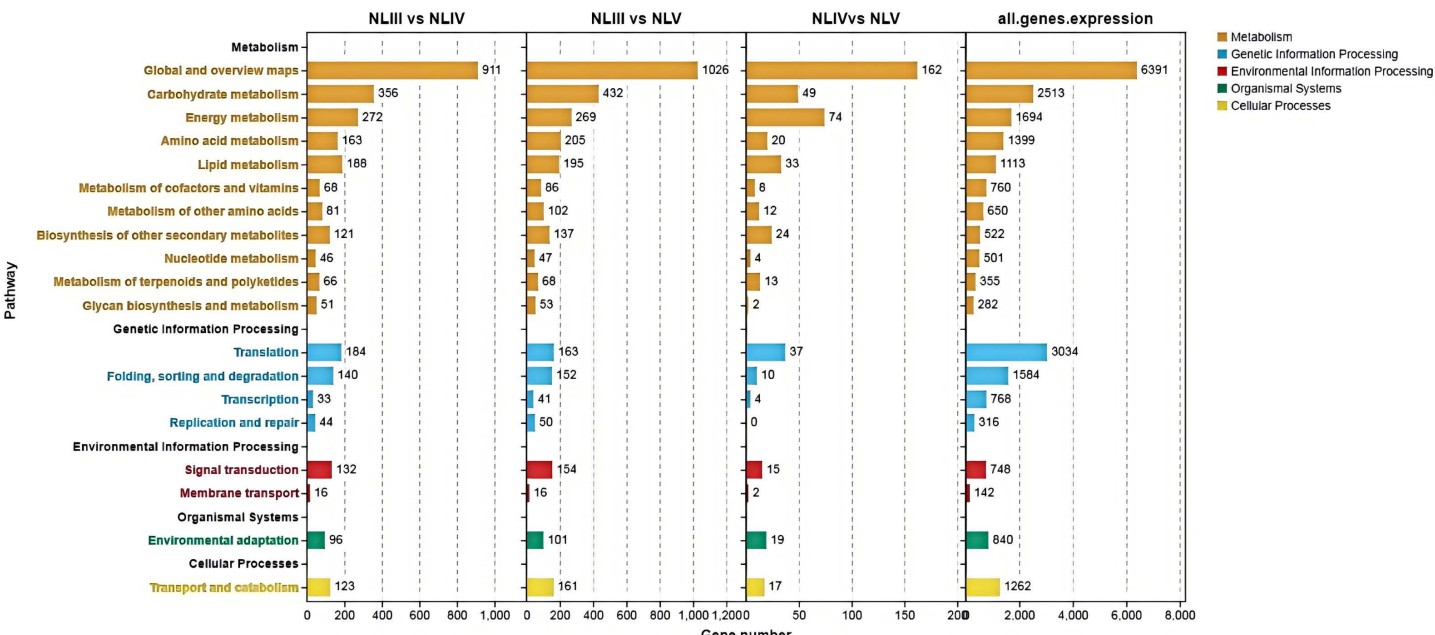

**Fig 8. KEGG pathway analysis of DEGs in each comparison of new leaves.**

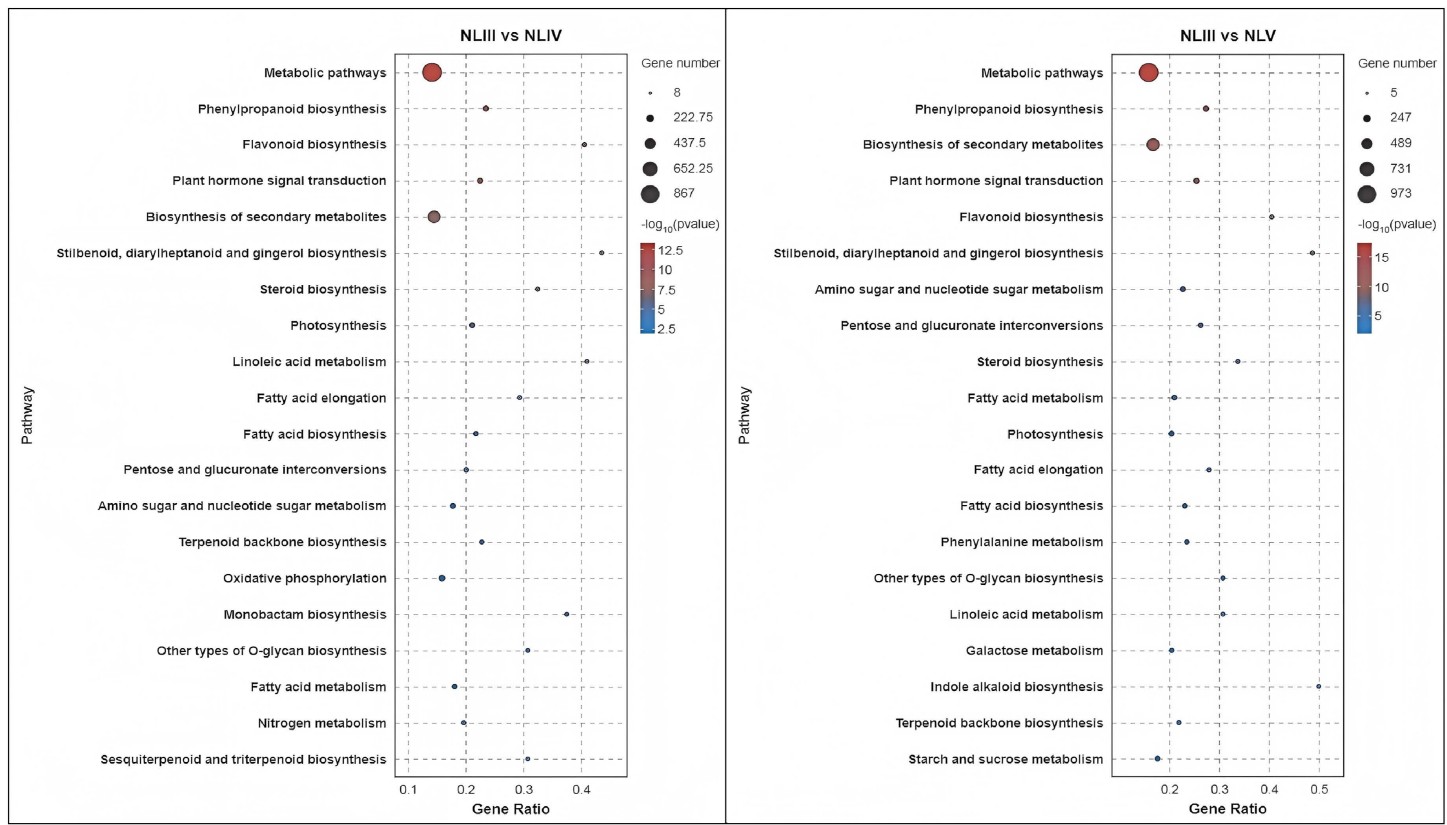

**Fig 9. Top 25 significance level KEGG pathway in each comparison of new leaves.**

Enrichment analysis revealed that the turquoise module was predominantly enriched for photosynthesis-related GO terms (e.g., photosystem II, photosynthetic electron transport) and KEGG pathways including "Photosynthesis - antenna proteins" and "Carbon fixation in photosynthetic organisms". This indicates that the coordinated downregulation of this core photosynthetic network is a key transcriptional feature of leaf functional maturation in *C. oleifera* (Supplementary S1 Fig).

## Dynamic regulation of photosynthesis and linked metabolic pathways

A detailed map of the photosynthesis pathway (ko00195) showed coordinated upregulation of genes encoding light-harvesting complexes (Lhca/b), photosystem I/II subunits (Psa/Psb), cytochrome b6/f complex, and ATP synthase from NLIII to NLV (Fig 10). Notably, 15 photosystem II-related genes and 10 photosystem I-related genes were upregulated by >2-fold in NLV compared to NLIII, aligning with the rise in chlorophyll *a*.

The enhancement of photosynthetic capacity was coupled with induction of Calvin cycle genes (e.g., RBCS, FB Pase, PRK) and downstream pathways including starch/sucrose metabolism and fatty acid biosynthesis. Specifically, key genes involved in triacylglycerol assembly (e.g., DGAT, PDAT) were upregulated at NLV, suggesting a metabolic link between photosynthetic output and oil precursor synthesis.

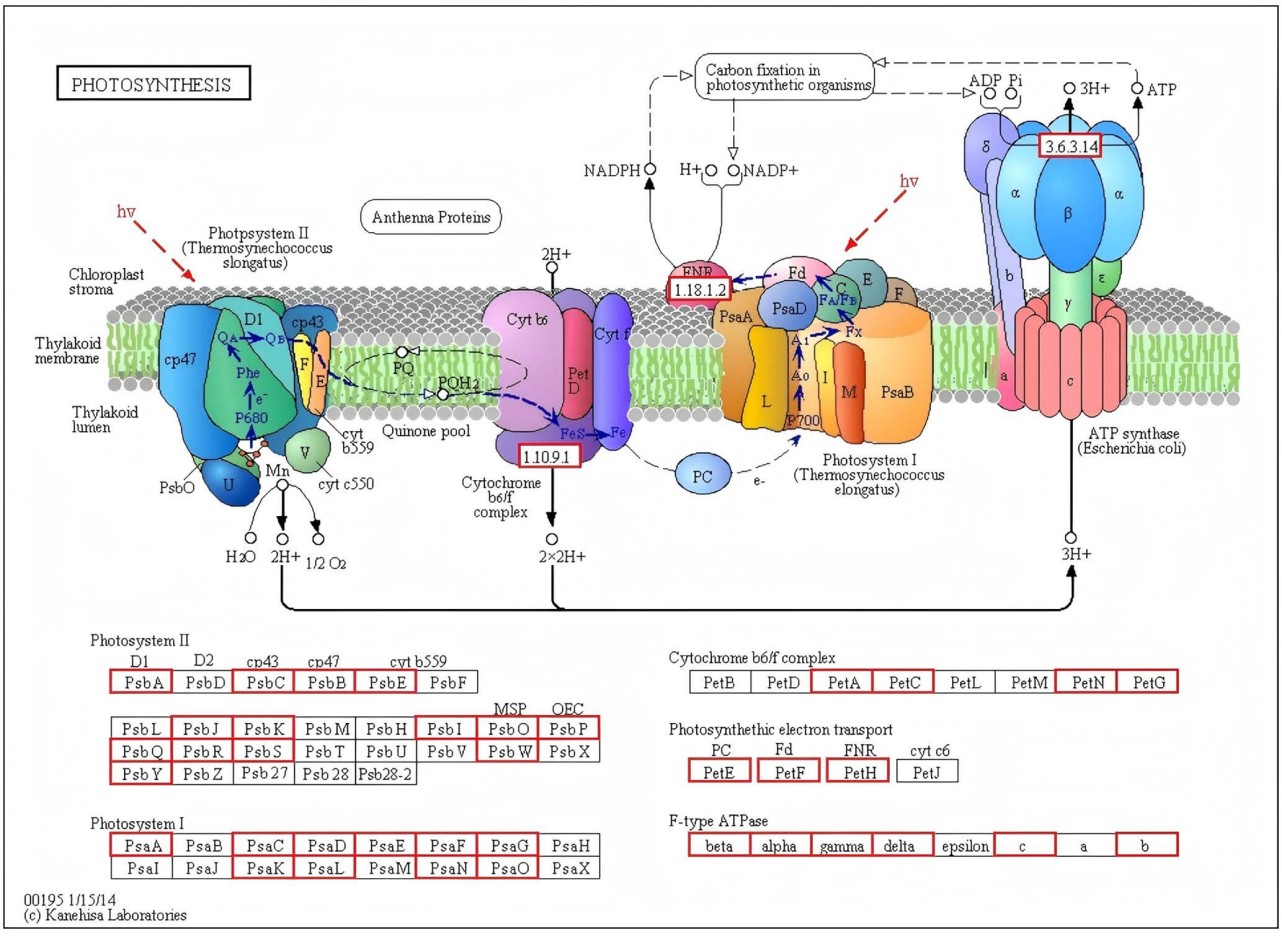

**Fig 10. DEGs involved in the Photosynthesis signal transduction pathway.**

## Validation of RNA-Seq data by qRT-PCR

To validate the relative expression levels from the transcript abundance estimation, A total of 12 differentially expressed genes were selected at random for PCR validation. The results demonstrated that the pattern of variation in DEG expression levels identified by qRT-PCR was consistent with the RNA-seq data (see Fig 11). This finding suggests that the RNA-seq data obtained in this study can be considered reliable.

## Discussion

To achieve their final shape and size, plant leaves must undergo three pivotal events: initiation, outgrowth, and expansion. Regardless of the final leaf shape, the incipient leaf initiates as a simple primordium from the shoot apical meristem (SAM) [21]. Once asymmetry is established, this leaf primordium undergoes subsequent elongation and partitioning into a

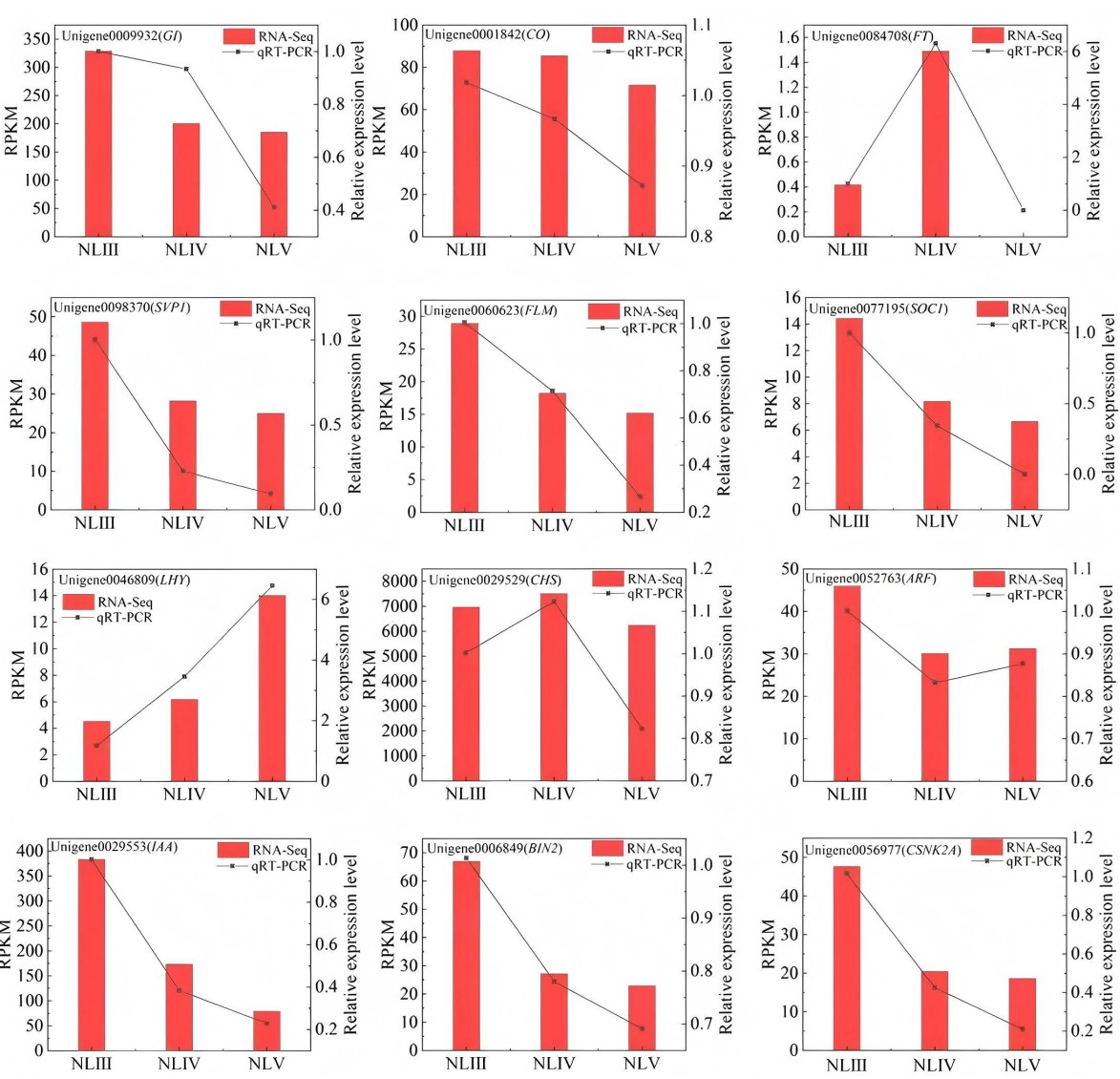

**Fig 11. The qRT-PCR validation for transcriptome data in new leaves.**

proximal petiole [22]. For this study, we aimed to examine the dynamic process of leaf development in L. chinense.In this experiment, we performed transcriptome sequencing on the fresh leaves of Changlin 53# *C. oleifera* at three periods: leaf spreading, growth and maturity. The results showed that the proportion of Q20 (sequencing error rate less than 1%) of the samples was 97.49%, and the average proportion of Q30 (sequencing error rate less than 0.1%) was 92.74%. The GC content of all samples ranged from 45.88% to 52.92%, with an average of 48.63%, indicating that the sequencing was relatively successful. To identify the differentially expressed genes (DEGs) in the new leaves, three comparison groups were created, i.e., NLIII vs. NLIV, NLIII vs. NLV, and NLIV vs. NLV. Subsequent analysis revealed that NLIII vs. NLIV, NLIII vs. NLV, and NLIV vs. NLV had 8225, 9443, and 981 differentially expressed genes (DEGs), respectively. Notably, NLIII vs. NLIV had more DEGs than NLIV vs. NLV. The gene expression dynamics were most active from leaf expansion to the growth stage, whereas the DEGs from the growth stage to the maturity stage were significantly reduced, which might be related to the gradual stabilization of leaf function.GO enrichment analysis indicated that the bioprocess category exhibited the highest percentage of metabolism- and cell-function-related DEGs. This finding aligns with the observed continuous increase in chlorophyll a content, suggesting that photosynthetic capacity undergoes enhancement during new leaf development. This phenomenon may promote leaf functional maturation through the regulation of carbon assimilation and energy metabolism pathways. This finding is analogous to the dynamic expression pattern of fatty acid metabolism genes during seed maturation in *C. oleifera*, suggesting the presence of commonalities in metabolic regulation across diverse organs [23].

## Stage-specific regulation of photosynthesis, hormone signaling, and secondary metabolism

Our transcriptome atlas reveals a clear temporal program during *C. oleifera* leaf development. The early phase (NLIII→NLIV) is characterized by activation of phenylpropanoid/flavonoid biosynthesis and stress-response pathways, likely driving cell wall fortification and antioxidant protection for expanding leaves [24]. The concurrent induction of JA and ABA signaling genes (LOX, AOS, PYL, SnRK2) suggests hormonal coordination in balancing growth with environmental acclimation, as reported in other perennials [25,26].

The transition to functional maturity (NLIV→NLV) is marked by a pronounced upregulation of photosynthetic apparatus genes and Calvin cycle enzymes, coinciding with the plateauing of chlorophyll *a* content. This indicates a shift from structural building to photosynthetic optimization. The parallel induction of fatty acid biosynthesis/elongation genes points to a metabolic channeling of photoassimilates toward lipid production, a feature potentially advantageous for an oilseed tree where leaves may contribute precursors for seed oil synthesis [27].

## Limitations and future perspectives

This study provides a comprehensive transcriptomic resource for *C. oleifera* leaf development but has limitations. First, while we identified key pathways and TFs, causal relationships require functional validation through genetic manipulation. Second, physiological parameters such as photosynthetic rate, stomatal conductance, and detailed lipid profiles were not measured; future studies should integrate multi-omics (transcriptome, proteome, metabolome) with physiological assays to establish direct functional links. Third, comparisons with other organs (e.g., seeds) remain indirect; systematic cross-organ transcriptomics would clarify shared versus unique regulatory modules. Fourth, WGCNA suggested co-expression modules but network inference from time-series data could further reveal temporal regulatory logic. Finally, expanding comparisons to more woody oil crops under controlled environments would help dissect species-specific adaptations.

Despite these limitations, our findings highlight stage-specific transcriptional reprogramming underlying leaf functional maturation in *C. oleifera*. The convergence of photosynthesis enhancement, phenylpropanoid-mediated strengthening, and lipid metabolic induction provides a framework for improving photosynthetic efficiency and stress resilience in this important oilseed tree.

## Comparative aspects with other woody oil crops

Unlike herbaceous oil crops (e.g., soybean, rapeseed), woody perennials like *C. oleifera*, oil palm, and tung tree (Vernicia fordii) invest heavily in long-lived leaves with sustained photosynthetic output. Oil palm shows strong coordination between leaf expansion and lipid metabolism gene expression during pinnate leaf maturation [28].Tung tree leaves highly express genes for fatty acid desaturation during peak photosynthesis [29].*C. oleifera* appears to share this trait but also shows a distinctive sustained upregulation of phenylpropanoid pathway genes into later stages, possibly reflecting its adaptation to subtropical forest understory conditions where UV and biotic stress resilience are critical. This may represent a unique evolutionary trade-off between defense and productivity in *C. oleifera*.

## Conclusion

Using integrative morphological, physiological, and transcriptomic analyses, we deciphered dynamic gene expression patterns during new leaf development in *C. oleifera* 'Changlin 53'. The early stage emphasized cell wall biogenesis and stress acclimation, while the late stage shifted toward photosynthetic optimization and lipid metabolism. Key TF families and co-expression modules associated with these transitions were identified. These findings reveal potential synergistic regulation among photosynthesis, secondary metabolism, and hormone signaling, offering candidate genes and pathways for molecular breeding of *C. oleifera* with improved light-use efficiency and yield potential.

## Supporting information

**S1 Table. Top 20 transcription factor families among DEGs.**
(XLSX)

**S2 Table. WGCNA module–trait relationships and hub gene network.**
(XLSX)

**S1 Fig. GO and KEGG enrichment results for the turquoise module.**
(JPG)

**S3 Table. qRT-PCR primer sequences.**
(XLSX)

## Author contributions

**Conceptualization:** Zongshun Zhou, Hongyan Guo.

**Data curation:** Zongshun Zhou, Chuansong Chen, Ying Jiang, Hang Wang.

**Formal analysis:** Zongshun Zhou, Chuansong Chen, Hang Wang.

**Funding acquisition:** Ying Jiang.

**Investigation:** Yikai Hua, Yisi Liu, Lixian Cao, Shuhua Wu.

**Methodology:** Zongshun Zhou, Chuansong Chen, Ying Jiang, Lixian Cao.

**Project administration:** Yikai Hua, Yisi Liu.

**Resources:** Zongshun Zhou, Hongyan Guo, Hang Wang.

**Supervision:** Shuhua Wu.

**Writing – original draft:** Zongshun Zhou, Ying Jiang.

**Writing – review & editing:** Zongshun Zhou.

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
