## [Decision Letter · Decision Letter 0]

30 Oct 2025

PONE-D-25-52961Analysis of leaf morphology development-related genes and photosynthetic metabolic pathways in the transcriptomes of new leaves of Tea-Oil tree ( Camellia oleifera  ‘changlin53’)PLOS ONE

Dear Dr. zhou,

Thank you for submitting your manuscript to PLOS ONE. After careful consideration, we feel that it has merit but does not fully meet PLOS ONE’s publication criteria as it currently stands. Therefore, we invite you to submit a revised version of the manuscript that addresses the points raised during the review process.

We look forward to receiving your revised manuscript.

Kind regards,

Mojtaba Kordrostami, Ph.D.

Academic Editor

PLOS ONE

Journal Requirements:

“none”

5. Please provide a complete Data Availability Statement in the submission form, ensuring you include all necessary access information or a reason for why you are unable to make your data freely accessible. If your research concerns only data provided within your submission, please write "All data are in the manuscript and/or supporting information files" as your Data Availability Statement.

7. Please ensure that you include a title page within your main document. You should list all authors and all affiliations as per our author instructions and clearly indicate the corresponding author.

Reviewer's Responses to Questions

**Comments to the Author**

1. Is the manuscript technically sound, and do the data support the conclusions?

Reviewer #1: Yes

Reviewer #2: Partly

Reviewer #3: Partly

Reviewer #4: Yes

2. Has the statistical analysis been performed appropriately and rigorously?

Reviewer #1: Yes

Reviewer #2: N/A

Reviewer #3: N/A

Reviewer #4: Yes

3. Have the authors made all data underlying the findings in their manuscript fully available?

Reviewer #1: Yes

Reviewer #2: No

Reviewer #3: Yes

Reviewer #4: No

4. Is the manuscript presented in an intelligible fashion and written in standard English?

Reviewer #1: Yes

Reviewer #2: No

Reviewer #3: No

Reviewer #4: No

5. Review Comments to the Author

Reviewer #1: This study combined morphological and anatomical observations with multi-stage transcriptome sequencing to analyze the key regulatory networks underlying leaf functional maturation in Camellia oleifera 'Changlin 53'. However, the manuscript contains multiple issues with figure presentation and description. There are still some issues that need be improved and revised.

1.In Figure 1, the leaf buds are shown without a scale bar, making it difficult for readers to assess their actual size. It is recommended to include an appropriate scale bar or indicate magnification to allow for accurate interpretation of the morphological features.

2.The results presented in Figures 2 and 3 are essentially part of a single logical dataset. It is recommended to merge them into one figure (as Figure 2) to improve the coherence of data presentation. In the corresponding text, the authors should refer precisely to the specific panels (e.g., “Figure 3a”) rather than using a broad reference such as “Figures 2–3” to ensure accurate correspondence between the narrative and the visual data.

3.In the “Study of new growth pattern” section the manuscript states:

“The results of Figures 2-2, 2-3, and 2-4 show that chlorophyll A content in new leaves during oil tea growth exhibited a similar pattern, indicating that chlorophyll A content can be used to evaluate oil tea growth.”

The reference to “Figures 2-2, 2-3, and 2-4” is unclear. I could not find a figure labelled “Fig. 2-2” in the file; it is ambiguous whether the authors mean “Figs. 2–4”, “Figs. 2-2/2-4”, or specific subpanels (e.g., Fig. 2a, Fig. 2b). Please carefully check and correct all figure citations in the text so that each in-text reference matches an existing figure or panel.

4.In the “Sequencing quality analysis” section, the description of the results should explicitly indicate that the data are derived from Table 2-1. However, the current table numbering (Table 2-1) is inconsistent with standard journal formatting. It is recommended to simplify it to “Table 1” (Tab. 1) and to ensure consistent numbering throughout the manuscript. Additionally, please make sure that all references to tables in the text correspond correctly to the actual table numbers.

5.In the Functional Annotation of Unigenes section, the scientific (Latin) names of species are not italicized. According to standard biological nomenclature rules, all genus and species names should be presented in italics (e.g., C. oleifera). Please carefully revise this section to ensure all Latin names are correctly formatted.

6.In the Sample relationship analysis section, the text refers to “(Figure 2-1)”; however, this figure does not appear to correspond to the numbering in the submitted manuscript. Please carefully verify all figure citations to ensure that each in-text reference accurately matches the correct figure or panel. Consistent and correct figure numbering is essential for clarity and to avoid confusion for readers.

7.There are two figures both labeled as “Figure 6” in the manuscript. Please carefully check the figure numbering throughout the manuscript to ensure that each figure has a unique number and that in-text citations correspond correctly to the figures.

Additionally, the legend of the second Figure 6 refers to panels “a” and “b”, but these subpanels are not present in the figure itself. Please verify that all figure panels are correctly displayed and that the legends accurately describe the figure content. Proper alignment between figures, subpanels, and captions is essential for clarity and accurate interpretation.

8.There are numerous instances of mismatched figure numbers and in-text citations throughout the manuscript. Please carefully verify all figures, subpanels, and references to ensure consistency and correct alignment.

9.In the Plant Growth and Sample Collection section, the coordinates are not correctly formatted.

10.In the Discussion section, “SAM19” appears, where “19” refers to a reference. According to standard journal conventions, the full reference should be provided when it is first mentioned, and subsequent mentions may use the abbreviation. Please carefully check the manuscript for similar instances where abbreviations or reference numbers are used at first mention, and ensure all follow proper formatting rules.

11.In the Methods section, the manuscript does not clearly indicate the developmental stage or sampling time corresponding to the samples labeled NLⅢ, NLⅣ, and NLⅤ. Please specify the exact stage or time point for each sample to ensure that the experimental design is fully reproducible and interpretable.

12.In section 4.6, the expression “2 −Ct” appears incorrectly formatted with respect to superscripts and subscripts. In standard qPCR notation, it should be written as “2−ΔΔCt ” or the appropriate formula intended. Please carefully check the entire manuscript for similar issues of superscript and subscript misusage to ensure all formulas and gene expression notations are correctly presented.

Reviewer #2: This study presented the first systematic elucidation of the molecular mechanisms underlying the leaves development in Camellia oleifera. They found that the enhancement of photosynthesis is closely coordinated with carbon and nitrogen metabolism during leaf maturation by using transcriptome sequencing. This result offers molecular insights for improving yield and quality. However, there are several limitations in the experimental design, results, and derivation of conclusions. The following points require further clarification or supplementation to strengthen the overall validity and impact of the study.

1.As for transcriptome data, although a number of DEGs were identified, key regulatory genes or core modules (for example, transcription factors, TFs) were not highlighted. WGCNA is helpful for in-depth analysis of the “photosynthesis–metabolism” regulatory network.

2.In Figures 7-10, the authors presented GO and KEGG enrichment analysis at the surface level. Visualizing key DEGs or gene–pathway association networks would improve interpretability.

3.Although numerous pathways (for example, PAL, 4CL, FLS) were presented, logically, it is unclear how different pathways (photosynthesis, hormones, secondary metabolism) change during different phases. The related contents should be added in the discussion section.

4. The authors should emphasize the comparison of C. oleifera leaf development and other woody oil crops (for example, Vernicia fordii, oil palm) in addition to tomato and then discuss whether C. oleifera exhibits unique photosynthetic regulation or metabolic advantages.

5. The text need to be edited by a native English speaker.

Reviewer #3: This manuscript investigates the gene expression dynamics and photosynthetic metabolic pathways during new leaf development in Camellia oleifera'Changlin53' by integrating morphological observations and transcriptomic analyses. The study reveals critical mechanisms underlying leaf functional maturation, offering theoretical insights for improving oil tea yield through breeding. However, significant revisions are required to address language, structural, and methodological issues before publication. Specifically, I have the following questions:

1. There are grammatical errors and unclear expressions in the Abstract and main text. For example, inconsistent usage of 'oil tea' and 'tea oil', and a spelling error in 'photosynthesis-mochromism'. It is recommended that the manuscript undergo a thorough revision by native English speakers or professional editors, with a focus on simplifying sentence structures and correcting grammatical inaccuracies;

2. The qPCR data, although mentioning three biological replicates, were not presented in the figures. It is recommended to supplement statistical analyses (e.g., ANOVA) to validate the reliability of the results;

3. The figure captions are inconsistent with the main text. For example, 'Figure 2-1' lacks a corresponding figure caption. 'Figure 1' is not cited in the text and should be either removed or properly referenced. For the figure image, it is recommended to add small icons to highlight key features;

4. The Discussion section is somewhat superficial and needs to be rewritten for greater depth and analytical rigor;

5. The manuscript lacks a description of the chlorophyll content determination methods;

6. Some pictures are suggested to be reorganized for presentation. The pictures are too simple and unaesthetic;

7. In Figure 8 (Photosynthetic Signaling Pathways), the annotations for gene upregulation/downregulation lack clarity, and there is no direct linkage between these molecular changes and physiological parameters (e.g., chlorophyll content, photosynthetic efficiency). Integration of multi-omics datasets (physiological and molecular) is strongly recommended to strengthen mechanistic interpretations and experimental validity.；

8. It is recommended to supplement the determination of photosynthetic parameters, as chlorophyll content may not necessarily correlate with photosynthetic efficiency.;

9. The conclusion section asserts that 'there are shared regulatory mechanisms with processes such as seed maturation', yet lacks comparative data across different organs. It is advisable to either refine the scope of conclusions or provide a critical discussion of the study's limitations in organ-specific regulatory networks;

10. The Results section lacks logical flow and is disorganized. It is recommended to reorganize the content following the sequence: morphological changes → transcriptomic dynamics → pathway analysis;

11. The association between ABA and JA signaling pathways and leaf stress resistance requires further experimental validation; additional data support is recommended;

12. The reference formatting is inconsistent, and citation information is incomplete in some entries. It is recommended to revise them in accordance with the target journal's guidelines;

14. qRT-PCR primer sequences are missing;

15. RNA-Seq library construction workflow is missing.

Reviewer #4: Overall Recommendation: Major Revision

I have gone through the manuscript entitled “Analysis of leaf morphology development-related genes and photosynthetic metabolic pathways in the transcriptomes of new leaves of Tea-Oil tree ( Camellia oleifera‘changlin53’)”. This manuscript investigates the molecular mechanisms of leaf development in the high-yielding oil-tea camellia ‘Changlin 53’ through transcriptomic analysis of three key developmental stages. The study identifies DEGs and proposes a valuable “photosynthesis-metabolism-hormones” synergistic model, providing a potential theoretical basis for breeding. The study was well-designed, and the key genetic data from the RNA-Seq analysis are reliable, a finding which was confirmed by qRT-PCR. The study has the potential to contribute valuable insights into the molecular mechanisms underlying leaf functional maturation in oil tea.

However, the manuscript in its current form requires significant improvement in language, data presentation, and clarity of descriptions before it can be considered for publication. Major concerns regarding data availability and figure completeness must be addressed.

Major Concerns

Language and Clarity:

The introduction of the research background in the Abstract section is too lengthy and needs to be condensed. The manuscript requires extensive English language editing by a native speaker or professional service. There are numerous grammatical errors, awkward phrasings, and unclear sentences throughout the text that hinder comprehension. For example, “new tip leaf spreading stage” may could be standardized to “leaf expansion stage”, and many sentences in the Results section are overly descriptive without clear biological context.

Data Availability Statement:

The Data Availability section is marked as “none”, and the text box states “The authors confirm that all data underlying the findings described are fully available without restriction”, but no location (e.g., repository name, accession number) is provided. This is a critical violation of PLOS ONE’s data policy. The raw RNA-Seq sequencing data must be deposited in a public repository (e.g., NCBI SRA, GEO) and the corresponding accession number(s) must be provided in the manuscript.

Figures and Tables:

Many figures referenced in the text (e.g., Figure 2-1, 2-2, 2-3, 2-4, 2-5, 2-8) are not included in the provided manuscript, or their labels are inconsistent (e.g., some figures are numbered as “Figure 1”, “Figure 2”, etc., while others use a “2-1” format). This makes it impossible to evaluate a significant portion of the results.

All figures must be thoroughly checked, re-labeled correctly and consistently, and provided with clear, detailed captions that allow the reader to understand the figure without referring to the text.

Methods Section:

The methodology lacks critical details necessary for reproducibility.

qRT-PCR: The sequences of the primers used for the 12 validated genes must be provided, ideally in a supplementary table.

RNA Quality: Specific values for RNA Integrity Numbers (RIN) or similar metrics should be reported.

DEG Criteria: The criteria for DEG analysis states |fold change|>2, which is equivalent to a log2FoldChange of |1|, but the text in section 2.7.1 states |log2(Fold Change)|>2. This discrepancy must be clarified.

Minor Concerns

Introduction:

The introduction is generally adequate but could be improved by more clearly stating the specific knowledge gap this study aims to fill, particularly for C. oleifera.

Results:

The description of the KEGG enrichment results (section 2.7.3) is repetitive and somewhat confusing. The pathways are listed multiple times. This section should be streamlined to clearly state the most significant findings for each comparison group.

The biological interpretation of the GO and KEGG results should be strengthened. For instance, what is the specific implication of enriched pathways like “Stilbenoid, diarylheptanoid and gingerol biosynthesis” in the context of leaf development?

Discussion:

The discussion effectively links the findings to existing literature but could be more focused. It should start by succinctly summarizing the key findings and then elaborate on their significance. The statement about revealing the “potential synergistic mechanism of photosynthesis-metabolism-hormones” should be more explicitly supported by the data presented.

I recommend a Major Revision. The authors are encouraged to address all points raised above thoroughly. With careful revision, particularly in providing the sequencing data and correcting all figures, this manuscript has the potential to be a valuable contribution to PLOS ONE.

6. PLOS authors have the option to publish the peer review history of their article (what does this mean?). If published, this will include your full peer review and any attached files.

Reviewer #1: No

Reviewer #2: No

Reviewer #3: No

Reviewer #4: No

---

## [Decision Letter · Decision Letter 1]

27 Jan 2026

PONE-D-25-52961R1Analysis of leaf morphology development-related genes and photosynthetic metabolic pathways in the transcriptomes of new leaves of Tea-Oil tree ( Camellia oleifera  ‘changlin53’)PLOS One

Dear Dr. zhou,

Thank you for submitting your manuscript to PLOS ONE. After careful consideration, we feel that it has merit but does not fully meet PLOS ONE’s publication criteria as it currently stands. Therefore, we invite you to submit a revised version of the manuscript that addresses the points raised during the review process.

We look forward to receiving your revised manuscript.

Kind regards,

Mojtaba Kordrostami, Ph.D.

Academic Editor

PLOS One

Journal Requirements:

Reviewers' comments:

Reviewer's Responses to Questions

**Comments to the Author**

1. If the authors have adequately addressed your comments raised in a previous round of review and you feel that this manuscript is now acceptable for publication, you may indicate that here to bypass the “Comments to the Author” section, enter your conflict of interest statement in the “Confidential to Editor” section, and submit your "Accept" recommendation.

Reviewer #1: All comments have been addressed

Reviewer #2: (No Response)

2. Is the manuscript technically sound, and do the data support the conclusions?

Reviewer #1: Yes

Reviewer #2: Partly

3. Has the statistical analysis been performed appropriately and rigorously?

Reviewer #1: Yes

Reviewer #2: (No Response)

4. Have the authors made all data underlying the findings in their manuscript fully available?

Reviewer #1: Yes

Reviewer #2: (No Response)

5. Is the manuscript presented in an intelligible fashion and written in standard English?

Reviewer #1: Yes

Reviewer #2: (No Response)

6. Review Comments to the Author

Reviewer #1: There are some issues that need be improved and minor revised.

1. Please note that after the first full introduction of the species as "Camellia oleifera" , its subsequent mentions in the text, figures, and tables should be abbreviated to "C. oleifera" for consistency and in accordance with standard botanical nomenclature conventions.

2. Please resolve all instances of the citation error "Error! Reference source not found" (e.g., in Plant Growth and Sample Collection, line 6). Ensure every citation is correctly linked to its reference list entry.

3. In the Introduction section (Paragraph 2, Line 9), there is a citation formatting error: "ecological restoration 11,1212". The reference number "12" appears to be duplicated. Please systematically check all in-text citations for similar repetitive or incorrect numbering and ensure they are correctly linked to the reference list.

4. Figure 2 contains unreadable or unknown content, as highlighted in the attached screenshot.

5. The manuscript contains sporadic Chinese characters (e.g., full-width punctuation such as “：”, in the caption of Figure 2 ) and instances of missing spaces between words. These formatting errors detract from the text's professionalism and readability. Please conduct a thorough, line-by-line review to correct these issues and ensure the manuscript adheres to standard English formatting conventions.

6. The caption for Figure 5 is incomplete.

7. In the section "KEGG enrichment of DEGs," the phrase "as illustrated in Figure 8" appears twice in close proximity, creating unnecessary redundancy and disrupting the narrative flow. Please revise the text to state the finding concisely and cite the figure only once, typically at the most logical point where the data is first summarized or In the Methods section, you state that qRT-PCR experiments were performed with three replicates. However, the corresponding Figure 11 does not display any error bars or measures of variability (e.g., standard deviation or standard error of the mean).

8. In the Discussion section, The phrase "SAM 23" is ambiguous. Clarify whether "23" is a referenceor part of a name ?

Reviewer #2: (No Response)

7. PLOS authors have the option to publish the peer review history of their article (what does this mean?). If published, this will include your full peer review and any attached files.

Reviewer #1: No

Reviewer #2: No

---

## [Author Response · Author response to Decision Letter 2]

24 Feb 2026

All modifications have been completed.

---

## [Decision Letter · Decision Letter 2]

30 Apr 2026

Analysis of leaf morphology development-related genes and photosynthetic metabolic pathways in the transcriptomes of new leaves of Tea-Oil tree ( Camellia oleifera  ‘changlin53’)

PONE-D-25-52961R2

Dear Dr. zhou,

We’re pleased to inform you that your manuscript has been judged scientifically suitable for publication and will be formally accepted for publication once it meets all outstanding technical requirements.

Kind regards,

Taimoor Hassan Farooq

Academic Editor

PLOS One

Additional Editor Comments (optional):

Reviewers' comments:

Reviewer's Responses to Questions

**Comments to the Author**

1. If the authors have adequately addressed your comments raised in a previous round of review and you feel that this manuscript is now acceptable for publication, you may indicate that here to bypass the “Comments to the Author” section, enter your conflict of interest statement in the “Confidential to Editor” section, and submit your "Accept" recommendation.

Reviewer #1: All comments have been addressed

Reviewer #2: All comments have been addressed

2. Is the manuscript technically sound, and do the data support the conclusions?

Reviewer #1: Yes

Reviewer #2: Yes

3. Has the statistical analysis been performed appropriately and rigorously?

Reviewer #1: Yes

Reviewer #2: Yes

4. Have the authors made all data underlying the findings in their manuscript fully available?

Reviewer #1: Yes

Reviewer #2: Yes

5. Is the manuscript presented in an intelligible fashion and written in standard English?

Reviewer #1: Yes

Reviewer #2: Yes

6. Review Comments to the Author

Reviewer #1: Camellia oleifera is an important woody edible oil tree species in China. This paper “Analysis of leaf morphology development-related genes an d photosynthetic metabolic pathways in the transcriptomes of new leaves of Tea-Oil tree (Camellia oleifera ‘changlin5 3’)” was good. I recommented this paper to accept.

Reviewer #2: (No Response)

7. PLOS authors have the option to publish the peer review history of their article (what does this mean?). If published, this will include your full peer review and any attached files.

Reviewer #1: No

Reviewer #2: No

---

## [Editor Report · Acceptance letter]

PONE-D-25-52961R2

PLOS One

Dear Dr. zhou,

I'm pleased to inform you that your manuscript has been deemed suitable for publication in PLOS One. Congratulations! Your manuscript is now being handed over to our production team.

Kind regards,

on behalf of

Taimoor Hassan Farooq

Academic Editor

PLOS One